# High Surface Area–Activated Carbon Production from Cow Manure Controlled by Heat Treatment Conditions

**Jung Eun Park \***, **Gi Bbum Lee** , **Ho Kim and Bum Ui Hong**

Center for Bio Resource, Institute for Advanced Engineering, Yongin-si 17180, Korea;
mnbbv21c@gmail.com (G.B.L.); hokim0505@gmail.com (H.K.); buhong@iae.re.kr (B.U.H.)
* Correspondence: jepark0123@gmail.com; Tel.: +82-10-5477-5785

**Abstract:** In this study, methods of adding value to cow manure were studied. Due to the properties of cow manure, activated carbon with a high surface area can only be produced by increasing the fixed carbon ratio and removing the ash content. Activated carbon was fabricated using five different treatments: (1) raw material–chemical activation, (2) raw material–hydrothermal carbonization–chemical activation, (3) raw material–hydrothermal carbonization–chemical activation–acid washing, (4) raw material–hydrothermal carbonization–heat treatment–chemical activation, and (5) raw material–hydrothermal carbonization–chemical activation–acid washing. The products then underwent proximate, elementary, and surface area analyses. In addition, changes in activated carbon properties depending on the heat treatment temperature (300, 500, 700 °C) and the applied chemical activator ratios (1:1–1:3) were examined. The results showed that the best heat treatment temperature was 300 °C, and the cow manure to chemical activator ratio was 1:2. The heat treatment stabilization process increases the fixed carbon ratio and the solid yield, and the acid wash process removes substances that restrain the increase in surface area. Therefore, activated carbon with a surface area of 1955 $m^2/g$ can be produced after the addition of heat treatment and an acid wash to the process. In addition, the adsorption properties of activated carbon with different heat treatment conditions were studied.

**Keywords:** activated carbon; chemical activation; cow manure; high surface area; pretreatment conditions





## 1. Introduction

Based on the farmland restoration system, most livestock manure is recovered to be used as solid/liquid manure for farmland fertilization. Dense livestock farming areas, where farmland nutrient demand exceeds the amount of fertilizer derived from livestock manure, are at risk for agricultural environmental pollution [1,2]. In particular, drinking water source pollution due to livestock manure is a rising social issue, as concentrated nitrogen and phosphorus from chemical fertilizers leach into lakes and rivers and cause eutrophication [3]. Therefore, identifying measures to treat cow manure has become urgent. The Contracting Parties to the London Convention agreed to phase out dumping of livestock manure into the ocean [4]. Since the ban on livestock manure ocean dumping, studies on livestock manure treatment and resource recovery have increased.

Livestock manure management systems that currently focus on farmland resource circulation and recovery must be transformed to shift the focus to agricultural and rural area resource circulation via these value-adding technologies to recover livestock manure as a resource [5].

Among projects that convert livestock manure into energy sources, research on developing combined technologies that produce biogas from solid livestock manure is lacking [6]. Anaerobic digestion technology for converting solid livestock manure is still in its early stage of development [7–9].

Although solid fuel production technologies for converting livestock manure into fuels are being developed, they are still far from commercialization due to a lack of technologies

for solid fuel production as a public resource and insufficient supplier security for pilot-scale solid fuel production [10,11]. Moreover, the solid fuel production industry currently faces various challenges due to economic restraints from excessive input energy demand during the production process.

Therefore, the currently existing livestock manure management system focus on solid/liquid manure purification with waste-to-energy conversion requires reformation [12,13]. It is necessary to develop a technology to convert livestock manure into a high value-added resource, and the model should comply with the future paradigm shifts in livestock manure management.

Generally, cow manure is composed of 14% hemicellulose, 15% cellulose, and 7% lignin [14]. However, cow manure in South Korea contains sawdust and rice straw and thus has an approximately 20% higher lignin content than other livestock byproducts [15,16]. This increases the potential value of cow manure as an activated carbon precursor. However, cow manure generated from small-scale cowsheds in South Korea contains several impurities (extraneous matter such as sand and inorganic substances within raw materials), such as soil particles, which impede conversion to a high value-added resource. Several studies have been conducted to address this issue (Table 1).

**Table 1.** Future technological development areas for adding value to livestock byproducts [5].

| Defined | Yield (%) | Properties |
|---|---|---|
| Compost [1] | 20–40 | (1) Maturation period: 3 months<br>(2) Odor generation: 588 $m^3$/ton<br>(3) Increasing environment load<br>(N, $P_2O_5$, $K_2O$) |
| Solid Refuse Fuel [2] | 20–30 | (1) Odor generation: 1198 $m^3$/ton<br>(2) High drying energy: 553 Mcal/ton |
| Activated Carbon [3] | 10–20 | (1) Low economic feasibility |

1: Depending on maturation periods, the reduction rate is about 60% based on raw materials. 2: Based on solid refuse fuel (heating value: 3500 kcal/kg). 3: Based on commercial activated carbon (surface area: 900~1100 $m^2$/g, selling price: USD 4.0).

Most activated carbon with high surface area in the market is produced in the United States and Japan, with unit prices 7.2 USD/kg-AC and 6.5 USD/kg-AC, respectively, which is approximately 5–6 times higher than Chinese products (1.2 USD/kg-AC) [17]. Therefore, research on cow manure as a replacement to an activated carbon precursor is in process, as ensuring a consistent supply of precursors is essential for achieving activated carbon independence.

In previous research, the research team produced activated carbon from cow manure using a chemical activation reaction and utilized it for adsorption [18]. The largest surface area of AC was 946 $m^2$/g, and the AC based on cow manure-derived adsorbent had a strong interaction due to its large pore properties and more oxygen-containing complex on the surface. The Council of Scientific and Industrial Research in India fabricated activated carbon from cow manure and used it to eliminate the heavy metal component Cr(VI) in a solution [19]. However, impurities such as soil particles restrained the utilization of the byproduct.

Generally, the surface area and pore size of activated carbon are formed by carbon consumption. However, activated carbon based on cow manure largely influence the yields [20]. To add value to cow manure, impurities must be effectively controlled.

In addition, to adsorb odorous gases, especially ammonia, it is necessary to obtain more adsorption data on activated carbon to elucidate the influence of the surface oxides on ammonia adsorption. In previous research, activated carbon pretreated by ozone treatment led to an increase in ammonia removal efficiency [21], and acidic surface oxides increased the adsorption capacity [22,23]. However, the relationship between the amount of oxyacid groups on the adsorbent surface and ammonia capacity has not yet been explained [24].

Therefore, this research investigates the removal of ammonia using an adsorbent, such as activated carbon.

In this study, the surface area of the activated carbon was compared and evaluated based on the augmentation processes of impurities to propose a process for the high surface area–activated carbon production from cow manure.

## 2. Materials and Methods

### 2.1. Raw Materials

The cow manure used in this study was collected from a cowshed in Anseong City (Gyeonggi Province, South Korea) and stored in Sudokwon Landfill Site (Incheon, South Korea). All raw materials were dried at 105 °C to obtain dried cow manure-dried (CM-D).

### 2.2. Processes

The ACs were prepared by five types of pathways (Cases 1, 2, 3, 4, and 5): chemical activation (CM-DC), hydrothermal carbonization–chemical activation (CM-DTC), hydrothermal carbonization–chemical activation–acid washing (CM-DTCA), hydrothermal carbonization–heat treatment–chemical activation (CM-DTHC), and hydrothermal carbonization–heat treatment–chemical activation–acid washing (CM-DTHCA) (Figure 1).

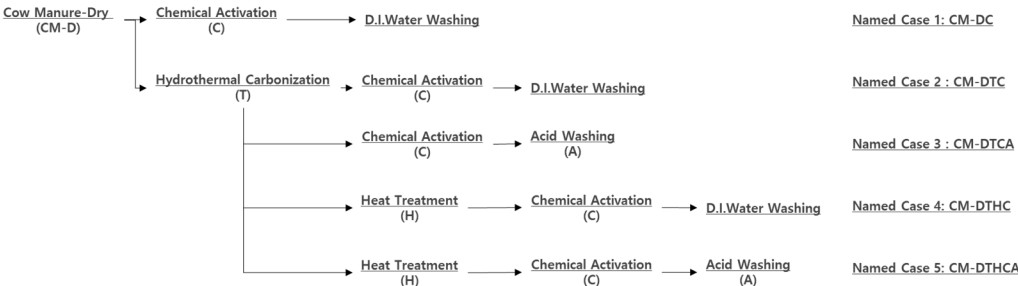

**Figure 1.** The various preparation methods (Hydrothermal carbonization, Heat treatment, Chemical activation, Acid washing) for activated carbon-based cow manure.

### 2.2.1. Hydrothermal Carbonization (HTC)

To increase the fixed carbon ratio content, the cow manure underwent hydrothermal carbonization (HTC) [25,26]. The reaction conditions followed for the HTC were based on results from a previous study [27]. Dried cow manure was used to control the amount of water content input: the cow manure to water weight ratio was 1:2. As the reactants were placed in the reactor, $N_2$ was added to eliminate internal gas. After purging, the reactor was closed, the temperature was maintained at 200 °C for one hour of HTC, and the reactor was run at 200 rpm during the process. The reactor was then cooled to 25 °C and 1 bar, and the product was available after the separation of solid material denoted as CM-DT.

### 2.2.2. Heat Treatment

To increase the fixed carbon ratio content, the cow manure was pretreated by heat. The temperatures for heat treatment were within the range of the chemical activation temperature. The samples were denoted as CM-DTH. The samples were placed in alumina crucibles and placed into a tubular furnace under $N_2$ flow. The $N_2$ was purged at 100 mL/min. The temperature was increased to 300, 500, and 700 °C at a rate of 5 °C/min and maintained for 3 h to obtain CM-DTH-300, CM-DTH-500, and CM-DTH-700, respectively.

### 2.2.3. Chemical Activation

The prepared samples were used for chemical activation, with potassium hydroxide (KOH, Samchun Chemical, Seoul, Korea, ACS reagent > 97%) as the activator [28]. The sample to chemical activator (KOH) ratios were 1:1–1:3. The reactants were loaded on an alumina boat in the tube reactor and were purged with $N_2$ before proceeding with the reaction. The temperature was increased to 850 °C at a rate of 5 °C/min and then

maintained for 3 h. Once the reaction was complete, the samples were treated with deionized water until the solution reached pH 7 and then they were dried at 105 °C for 24 h (denoted as CM-DC and CM-DHC).

### 2.2.4. Acid Wash

Acid wash using inorganic acid was used for the acid treatment by phosphoric acid ($H_3PO_4$, Sigma-Aldrich, St. Louis, MO, USA, ACS reagent > 85% in $H_2O$) [29]. Ten milliliters of acid was applied to the 5 g sample and, after 1 h, the sample was washed with deionized water until the pH of the solution reached pH 7. The sample was then dried at 105 °C for 24 h.

### 2.3. Characterization

For the proximate analysis, dried samples were placed in a furnace (Daeheung Sci., Incheon, Korea, DF-4S) and heated at 950 °C for 7 min and then at 750 °C for 10 h. The ash, volatile matter, and fixed carbon content within the surface area were measured as weight percentages. An elemental analysis (EA) was conducted using an elemental analyzer (Flash EA 1112, Thermo Scientific, Milan, Italy) following the procedures described in a previous study [28]. The elemental content of carbon (C), hydrogen (H), oxygen (O), nitrogen (N), and sulfur (S) was determined. The surface area of the sample was analyzed using the Brunauer–Emmett–Teller (BET) method based on the $N_2$ adsorption at −196 °C using an adsorption analyzer (ASAP 2020, Micrometrics, Norcross, GA, USA). The pore sizes of the samples were calculated according to the Barrett Joyner and Halenda method. Before obtaining the $N_2$ isotherms, the samples were outgassed at 350 °C to a constant vacuum (P/Po = 2 μm Hg) for 6 h. The materials adsorbed on the samples were analyzed using X-ray fluorescence (XRF-1800, Shimadzu, Japan). The contents of the carbon functional group in the samples were confirmed via X-ray photoelectron spectroscopy (PHI-5000 VersaProbe, Ulvac-PHI, Kanagawa, Japan). To investigate surface morphologies of activated carbons with/without chemical activation by KOH, field-emission scanning electron microscopy (S-4300, Hitachi, Tokyo, Japan) was used. In addition, the thermal properties of the raw material and the carbonized sample were confirmed using a thermogravimetric analyzer (DTG-60M, Shimadzum, Japan).

### 2.4. Adsorption Ability Test

An ammonia adsorption capacity test was performed on the prepared activated carbon as shown in Figure 2. The ammonia was purged with $N_2$, and the adsorption capacity test proceeded at 800 ppm. The total flux was 2 L/min, and an ammonia gas analyzer (Multirae pro, Honeywell, Charlotte, NC, USA) was used to measure the ammonia concentration. Ammonia concentration was considered fully saturated ($C/C_0 > 0.95$) when the discharge concentration was equal to the initially injected concentration. The measured adsorption capacity was expressed in terms of adsorbent unit weight.

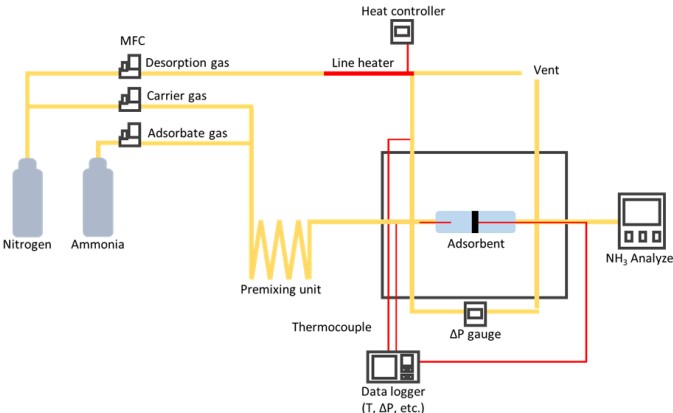

**Figure 2.** Schematic illustration of ammonia adsorption system.

### 3. Results

As discussed in introduction, various pretreatment methods were conducted to determine efficient processes for the activated carbon with high surface area. However, the physical properties of the dried cow manure (CM-D) differed depending on the location. The results of proximate analysis and ultimate analysis of CM-D are shown in Table 2.

**Table 2.** Proximate and ultimate analysis of CM-D.

| | Proximate Analysis (%)—Dry Basis | | | | Ultimate Analysis (%) | | | | |
|---|---|---|---|---|---|---|---|---|---|
| | Moisture | Volatile | Fixed Carbon | Ash | C | H | O | N | S |
| CM-D | 1.1 | 56.6 | 17.8 | 24.5 | 42.6 | 4.9 | 51.8 | 0.9 | 0.0 |

The proximate analysis of the cow manure supply showed that the fixed carbon, volatile matter, and ash contents were 17.8%, 56.6%, and 24.5%, respectively. From the ultimate analysis of CM-D, the contents of C, H, O, and N were 42.6, 4.8, 29.0, and 0.9%.

The composition of CM-D differs by location. In the USA [25], CM-D is 44.8% carbon and 9.4% fixed carbon, whereas in China [16], proximate and ultimate analyses of cow manure revealed 35.0% carbon and 12.4% fixed carbon content. The maximum and minimum fixed carbon content of biomass have been recorded as 20.4% and 16.0%, respectively [30]. Generally, biomass is used as a precursor of activated carbon, but it is mainly used as solid fuel due to excessive volatiles, which impede the growth of the surface area.

#### 3.1. Case 1: CM-DC

The results of the proximate and ultimate analyses of CM-DC are shown in Table 3. The CM-DC were prepared by chemical activation with various agent ratios on CM-D.

**Table 3.** Physical properties and the results of chemical activation on CM-DC.

| Sample to KOH Ratio(*w*/*w*) | Proximate Properties (%)—Dry Basis | | | | Solid Yield (%) | Surface Area (m²/g) |
|---|---|---|---|---|---|---|
| | Moisture | Volatiles | Fixed Carbon | Ash | | |
| 1:1 | 1.8 | 13.1 | 34.4 | 50.7 | 32.5 | 541 |
| 1:2 | 2.1 | 16.6 | 24.9 | 56.4 | 14.5 | 763 |
| 1:3 | 2.5 | 17.0 | 20.2 | 60.3 | 8.8 | 642 |

The surface area of the activated carbon was 540–763 m²/g. The ideal sample to KOH ratio was 1:2, achieving the highest surface area, whereas the surface area collapsed under the conditions of 1:3 or more, as found in previous research [29]. Solid yields were calculated the using the following equation and compared.

Solid Yield (%) = (Output solid as activated carbon)/(Input solid as precursor) × 100

The solid yields were in the range 8–32% due to the low fixed carbon content, and the solid yield tended to decrease with an increase in surface area. Therefore, they may be potentially utilized as substandard activated carbon, as the fixed carbon ratio must be increased to produce high surface area. Chemical activation of cow manure resulted in a 15% yield, the lowest of all treatments. Therefore, direct chemical activation of raw materials has disadvantages of very low yield with low surface area.

#### 3.2. Case 2: CM-DTC

To increase the fixed carbon content of CM-D, HTC process needed. Hydrothermal carbonized solid was dried at 105 °C for 24 h. From the results of HTC, the fixed carbon

content of CM-DT was increased by about 36% compared to that of CM-D, as shown in Table 4.

**Table 4.** The proximate analysis and ultimate analysis of CM-DT.

| | Proximate Properties (%)—Dry Basis | | | | Ultimate Analysis (%) | | | | | Surface Area (m²/g) |
|---|---|---|---|---|---|---|---|---|---|---|
| | Moisture | Volatiles | Fixed Carbon | Ash | C | H | O | N | S | |
| CM-DT | 0.2 | 57.3 | 24.2 | 18.4 | 45.6 | 3.6 | 49.3 | 0.8 | 0.6 | 9.0 |

CM-DTC were prepared by chemical activation with various agent ratios on dried CM-DT. The proximate properties of CM-DTC are shown in Table 5. The surface area of CM-DTC was generally higher than that of CM-DC. Specifically, a sample to KOH ratio ratio of 1:2 showed a 91% increase in surface area. The fixed carbon content of CM-DT was 36% higher than that of CM-D and that of CM-DTC was generally higher than that of CM-DC, except for a 1:1 sample to KOH ratio. In addition, CM-DTC had a lower ash content than CM-DC. The increase in surface area was significantly higher than the increase in the fixed carbon ratio.

**Table 5.** The proximate analysis, yield, and surface area of CM-DTC.

| Sample to KOH Ratio (*w/w*) | Proximate Properties (%)—Dry Basis | | | | Solid Yield (%) | Surface Area (m²/g) |
|---|---|---|---|---|---|---|
| | Moisture | Volatiles | Fixed Carbon | Ash | | |
| 1:1 | 2.0 | 17.3 | 25.5 | 55.3 | 27.3 | 1038 |
| 1:2 | 1.8 | 18.4 | 32.4 | 47.5 | 21.5 | 1110 |
| 1:3 | 2.2 | 15.8 | 40.5 | 41.4 | 25.2 | 1034 |

The results of a previous study about the acid treatment effect found that the ash content decreased [29]. However, if the concentration of acid was excessive or acid treatment is carried out before the carbon structure was stable, the collapse of the carbon structure progressed, and it acted as a hindrance to the development of activated carbon. [29,31] Otherwise, the better acid treatment conditions led to efficient acid removal.

The result of a previous study claimed that volatile matter content to ash content ratio contributed to the increase in surface area when washing with phosphoric acid instead of water, following the cow manure-(HTC)–chemical activation treatment, as shown in Table 6 [29]. From the results, the surface area of CM-DTCA was higher than CM-DTC and decreased the ash content. However, the CM-DTCA with the 1:1 sample to KOH ratio was shown to have the highest surface area. This means that when acid washing was performed under conditions with a weak carbon bonding force, the specific surface area increase was not large under the conditions of 1:2 and 1:3 sample to KOH ratios in which an excess amount of the activator was used. Therefore, the lower the activator ratio, the higher the content of ash physically bound to the surface, and higher the activator ratio, the lower the specific surface area growth rate.

**Table 6.** The physical properties and surface area of CD-DTCA.

| Sample to KOH Ratio (*w/w*) | Proximate Properties (%)—Dry Basis | | | | Surface Area (m²/g) | Surface Area Growth Rate (%) * |
|---|---|---|---|---|---|---|
| | Moisture | Volatiles | Fixed Carbon | Ash | | |
| 1:1 | 1.5 | 15.1 | 70.2 | 13.2 | 1589 | 53 |
| 1:2 | 0.8 | 19.8 | 56.4 | 21.3 | 1362 | 23 |
| 1:3 | 0.2 | 27.2 | 45.7 | 27.2 | 1118 | 8 |

* Surface area grow rate = surface area of CD-DTCA/CD-DTC.

### 3.3. Cow Manure–(HTC + Heat Treatment)–Chemical Activation

Stabilization through heat treatment was performed, and an increase in fixed carbon ratio and surface area of CM-DT was expected, as HTC only resulted in a 10% (35.4% of CM-DT—24.2% of CM-D) increased grade in fixed carbon ratio. Through thermogravimetric analysis shown in Figure 3, it can be confirmed that the weight loss decreased while the fixed carbon ratio increased under the influence of heat treatment at 300 °C.

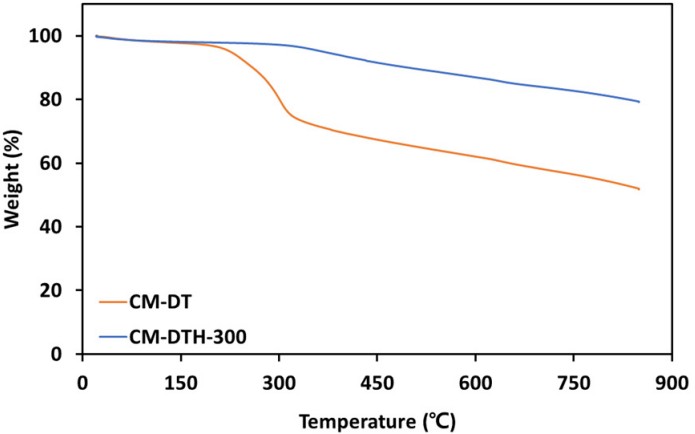

**Figure 3.** TGA thermograph for observation of heat treatment effect.

To control the fixed carbon content, cow manure was heat-treated at 300–700 °C. The physical properties of CM-DTH and CM-DTHC with different pre-heat temperatures are shown in Table 7 and Figure 4.

In the case of CM-DTH, the fixed carbon content generally increased with increasing pretreatment temperature. In addition, the volatile matter of CM-DTHs decreased with increasing temperature and the contnet of volatile matters of CM-DTHs are higher than CM-DT. However, CM-DTHs had low surface areas, even with hydrothermal and pre-heat treatments. In the case of CM-DTHC-700, the volatile content was increased with the activator ratio. There is a possiblity of collapse of the carbon structuree due to the high pretreatment temperature.

**Table 7.** The proximate properties of CM-DTH and CM-DTHC.

| Pre-Heat Temperature (°C) | Sample to KOH Ratio (*w/w*) | Proximate Properties (%)—Dry Basis | | | | Surface Area (m²/g) |
|---|---|---|---|---|---|---|
| | | Moisture | Volatiles | Fixed Carbon | Ash | |
| 300 | 1:0 | 0.6 | 39.0 | 35.4 | 26.2 | 14 |
| | 1:1 | 2.0 | 17.3 | 25.5 | 55.3 | 1216 |
| | 1:2 | 1.8 | 18.4 | 32.4 | 47.5 | 1330 |
| | 1:3 | 2.2 | 15.8 | 40.5 | 41.4 | 1272 |
| 500 | 1:0 | 0.5 | 18.0 | 48.2 | 34.3 | 14 |
| | 1:1 | 2.2 | 13.1 | 46.9 | 37.7 | 717 |
| | 1:2 | 4.9 | 17.3 | 37.0 | 40.8 | 1204 |
| | 1:3 | 3.0 | 13.5 | 40.9 | 42.7 | 1214 |
| 700 | 1:0 | 1.0 | 6.6 | 50.9 | 41.4 | 28 |
| | 1:1 | 2.0 | 18.1 | 46.1 | 33.8 | 559 |
| | 1:2 | 2.7 | 16.8 | 39.3 | 41.2 | 873 |
| | 1:3 | 2.1 | 16.4 | 41.5 | 40.1 | 830 |

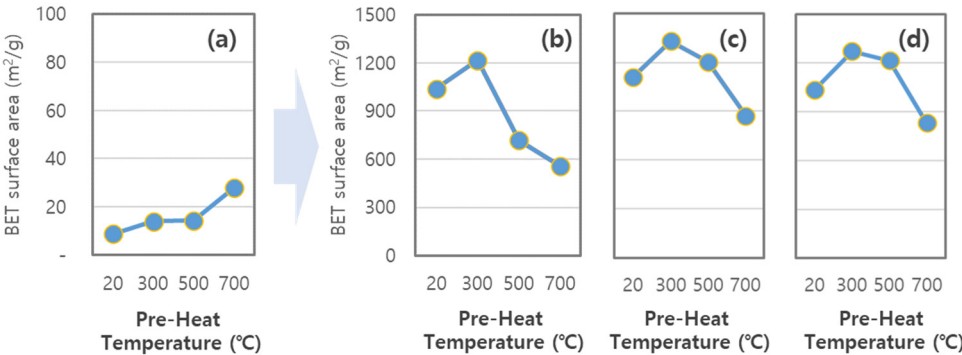

**Figure 4.** Relationship between the surface area and Pre-Heat temperatures by (**a**) CM-DTH and CM-DTHC with sample to KOH ratio (**b**) 1:1, (**c**) 1:2, and (**d**) 1:3.

During the chemical activation process, the surface area of the untreated livestock manure was approximately 1100 m$^2$/g, whereas the surface areas of manure with heat treatment at 300, 500, and 700 °C were 1330, 1204, and 873 m$^2$/g, respectively. Therefore, the surface area was readily developed at 300 °C, where pyrolysis did not occur. Although high-temperature heat treatment may increase the fixed carbon ratio, the surface area tends to decrease. Therefore, the ideal temperature for cow manure heat treatment is 300 °C, and the ideal cow manure to chemical activator ratio is 1:2. Nonetheless, the maximum surface area of 1330 m$^2$/g requires an additional step to become a high surface area–activated carbon.

As shown in Figure 5, we confirmed that a few micropores developed in the raw material, but the development of pores was hardly achieved. In the activated sample, the additional carbon consumption occurred as the amount of the chemical agent increased, and the size of the pores also increased proportionally. In the case of sample (d), mesopores or macropores mainly developed despite the increase in the number of cracks. It appears that the specific surface area is lowered due to the decrease in the ratio of micropores.

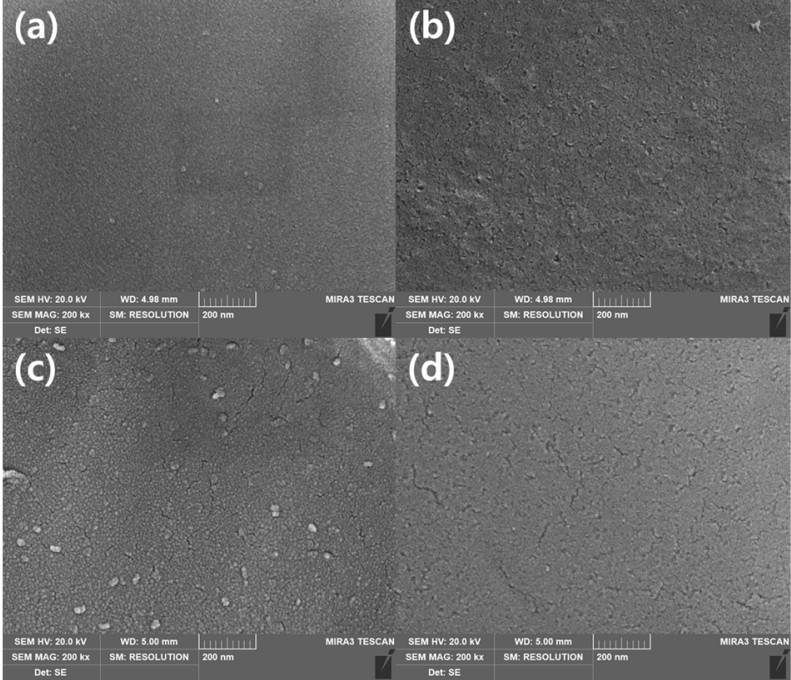

**Figure 5.** The scanning electron microscopy (SEM) of activated carbon: (**a**) CM-DTH, (**b**) CM-DTHC-300 (1:1), (**c**) CM-DTHC-300 (1:2), (**d**) CM-DTHC-300 (1:3).

Figure 6 shows the relationship between surface area and the ratio of fixed carbon to ash. Generally, the surface area increases with decreasing ash content, and a reduction in ash content is essential to improve the surface area [32]. In the case of CM-DTHC-300, the trend of surface area with an increasing ratio of fixed carbon to ash was to first increase and then to decrease. Therefore, it is hard to correlate the surface area of activated carbon based on cow manure to the ratio of fixed carbon to ash from this research. Even when the ratio of fixed carbon to ash was greater than or equal to 1, it was inversely proportional to the surface area. Nevertheless, the CM-DTHC-300 showed the highest surface area and thus had the best potential as high surface area–activated carbon.

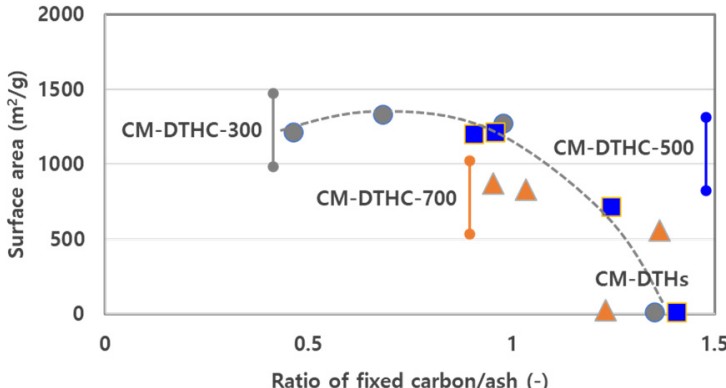

**Figure 6.** Relationship between fixed carbon/ash vs. surface area with different pre-heat treatment temperatures: 300 (circle), 500 (square), and 700 °C (triangle).

The oxygen functional group form and content affect the development of pores in the surface area. C-O and C=O formed at the center of the carbon lattice easily affect gas pocket development, whereas the entire carbon lattice volatilizes with the O-C=O form at the end of the carbon lattice and does not affect the pore development.

To compare the carbon functional group of CM-DTHCs, XPS analysis was performed, and the results are shown in Figure 7.

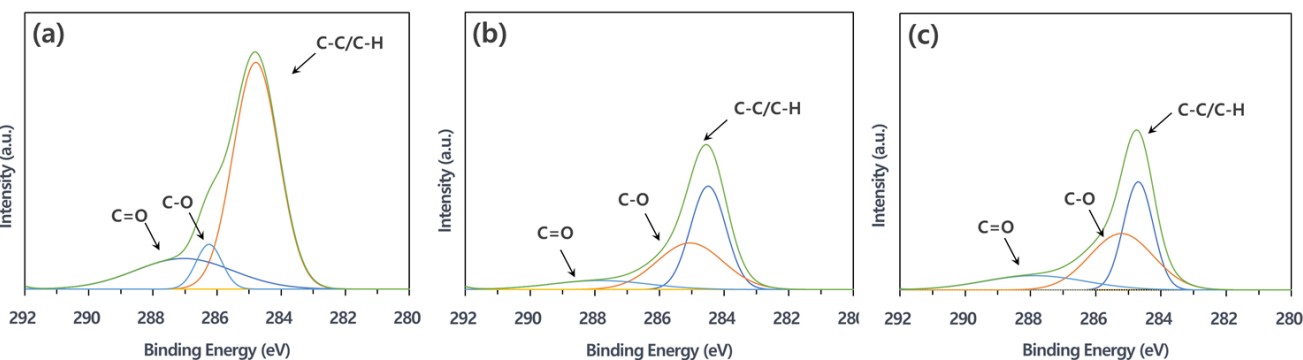

**Figure 7.** XPS spectrum of $C_{1s}$ for (**a**) CM-DTHC-300, (**b**) CM-DTHC-500, (**c**) CM-DTHC-700.

CM-DTH-300 did not decompose the carboxyl group through the C=O peak and showed a higher carbon content than other cases, as shown in Table 8. In CM-DTH-500 and CM-DTH-700, as the carboxyl group was decomposed by high temperature, the ratio of the phenol group decomposed at a relatively high temperature increased. In addition, the carbon lattice structure collapsed as the carbon content rapidly lowered. The increase in the ratio of C=O in CM-DTHC-700 can be attributed to the carbonyl group.

**Table 8.** The result of C1s XPS spectra of CM-DTHCs.

| Area (%) | CM-DTHC-300 | CM-DTHC-500 | CM-DTHC-700 |
|---|---|---|---|
| C-C | 62.7 | 48.1 | 38.7 |
| C-O | 9.5 | 40.3 | 43.4 |
| C=O | 27.8 | 11.7 | 17.9 |
| The fraction of C=O/(C-C+C-O) | 0.39 | 0.13 | 0.22 |

### 3.4. Cow Manure–(HTC + Heat Treatment)–Chemical Activation–Acid Wash

To determine the effect of acid washing of heat-treated samples at 300 °C, ash content and component analyses of the ash according to the presence or absence of acid washing were carried out, as shown in Figure 8.

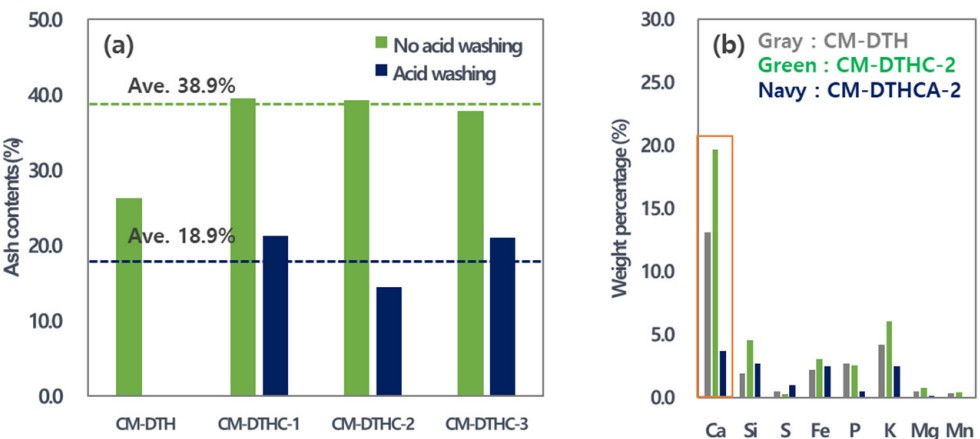

**Figure 8.** (**a**) Ash contents and (**b**) ash compositions of CM-DTHC with/without acid washing. All samples are based on CM-DTH-300.

As shown in Figure 8, the ash content of CM-DTHCA decreased by about 38.9% or more compared to CM-DTHC, regardless of the sample to KOH ratio. The surface area increased by about 8% under 1:1 and 1:3 sample to KOH ratio conditions, to 1325 and 1408 m$^2$/g, respectively. The high surface area of 1955 m$^2$/g was measured with a sample to KOH ratio of 1:2.

The results show that acid washing eliminates numerous ash components. The XRF analysis showed that the component of Ca significantly decreased.

Despite adding the HTC and heat treatment processes and proceeding with chemical activation to increase fixed carbon content, the surface area value remained at a suboptimal level. Thus, an acid wash process following the above processes was added to eliminate the ash content and produce a high surface area–activated carbon. Consequently, with a 1:2 sample to KOH ratio, activated carbon of approximately 1955 m$^2$/g was fabricated. It is considered that fixed carbon increase and ash content ratio (achieved through acid wash) affects the surface area of activated carbon.

However, increasing the number of processes resulted in a yield reduction and an increase in processing costs. Nonetheless, high surface area–activated carbon production using cow manure requires heat treatment and acid wash, as shown in Figure 9.

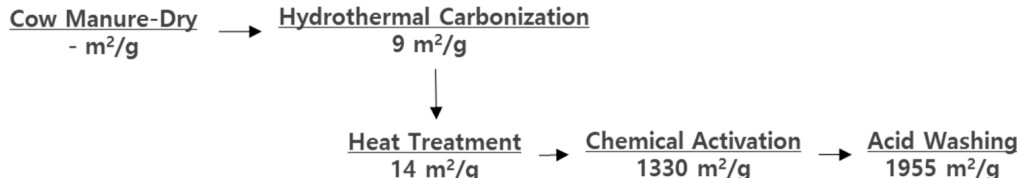

**Figure 9.** The suggested pathway to increase the surface area of the precursor of activated carbon.

### 3.5. Ammonia Adsorption Test

Generally, ammonia gas is notoriously hard to treat, and many researchers investigated how to increase the adsorption capacity [33]. To adsorb ammonia components, the low-cost abundant material of activated carbon is regarded as an effective approach. Activated carbon produced by our method underwent an ammonia adsorption test, and the results are shown in Figure 10.

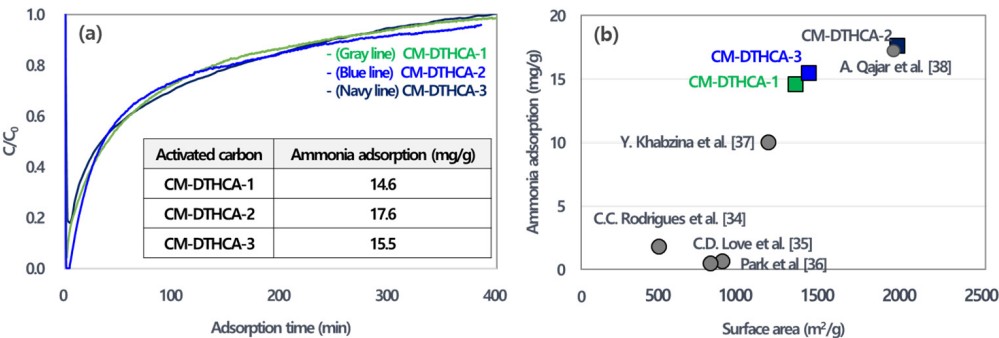

**Figure 10.** (**a**) Breakthrough curve of CM-DTHCAs (Co = 800 ppm; M = 1.0 g and $N_2$ = 2 L/min) and (**b**) relationship between surface area and ammonia adsorption of this research and previous research [34–38].

As shown in Figure 10a, the ammonia breakthrough curves were similar to sigmoidal-type mathematical expressions [34]. Rodrigues et al. noted the ammonia adsorption capacity is inversely proportional to temperature, and the adsorption pattern is related to the physical adsorption [34]. To increase the adsorption, Love and Park's research group studied ozone-treated activated carbon for increasing acid functional groups, such as carbonyl and ether groups [35,36]. Previous research determined that the adsorption process of ammonia onto activated carbon follows the pseudo-second order model [39]. The pseudo-second order kinetic model suggests that chemical adsorption is the dominant mechanism. The ammonia adsorbs on Lewis acid site at porous structure. Therefore, CM-DTHCA-2 showed the highest adsorption capacity due to the highest surface area and fraction of carbonyl groups.

Figure 10b shows the relationship between surface area and ammonia adsorption capacity [34–38]. The ammonia adsorption capacity of CM-DTHCA was similar to that determined in previous research (surface area: 1941 $m^2$/g with nitric acid treatment) [38]. From the results, the improvement of ammonia adsorption capacity of activated carbon was greatly affected by the surface area and acid functional groups.

In this study, the potential of converting cow manure to a high value-added product was evaluated, and it was determined that it is possible to produce high surface area –activated carbon from cow manure, with performances similar to those of commercial counterparts. Further research for commercialization will be conducted in future studies.

### 4. Conclusions

This study evaluated the potential of cow manure as a precursor to high surface area–activated carbon and compared its performance to commercial counterparts using the ammonia adsorption test. Consequently, direct chemical activation of cow manure with a low fixed carbon ratio resulted in low surface area and solid yield. Additionally, stabilization at high temperatures with unstable carbon bonds decreased the surface area and solid yield. The surface area and solid yield increased only when carbon bonds were stabilized before chemical activation. However, excessive input of chemical activators when carbon bonds of carbon precursors are weak decreases the surface area and solid yield. The ideal heat treatment temperature for stabilization was determined to be 300 °C, and the ideal cow manure to chemical activator ratio was 1:2. To produce high surface

area–activated carbon, at least 40% of the ash content in the manure must be eliminated through acid wash following chemical activation.

**Author Contributions:** Methodology and formal analysis, J.E.P. and G.B.L.; writing—original draft preparation and writing—review and editing, J.E.P. and G.B.L.; project administration, B.U.H.; funding acquisition, H.K. All authors have read and agreed to the published version of the manuscript.

**Funding:** This subject was supported by Korea Environmental Industry & Technology Institute (KEITI) as "Development of odor removal technology for treatment capacities (No.RE202101629)".

**Institutional Review Board Statement:** Not applicable.

**Informed Consent Statement:** Not applicable.

**Data Availability Statement:** Not applicable.

**Conflicts of Interest:** The authors declare no conflict of interest.

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
