# Peer review of "High Surface Area–Activated Carbon Production from Cow Manure Controlled by Heat Treatment Conditions"

_processes, doi:10.3390/pr10071282_

Round 1
Reviewer 1 Report
The introduction section lacks the main points that the literature has already reported on methodologies for the production of activated carbon presented in the manuscript;
There is no discussion of similar adsorbents reported in the literature in the introduction. There is only mention of a previous work published by the authors;
In section 2.1, please insert the geographic coordinates referring to the sample collection;
The figure caption of figure 1 is impressive. Please correct;
Please insert the discussion of table 2, comparing with the literature;
Correct "optimum" to "better (or similar)". Since there is no statistical analysis to determine the optimal point of the analyses. Also, none of the results are presented in duplicate/triplicate;
Please insert a reference to the information presented in lines 219-221;
Insert SEM and TGA analyzes of activated carbon obtained in the best condition;
The manuscript as a whole does not present a comparison of the analyzes with the literature, reducing its quality, despite being a work with a lot of potential;
Authors must review the manuscript for formal English before resubmitting the work, there are several spelling and concordance errors.
Author Response
Responses to Referee 1
General Comment: The introduction section lacks the main points that the literature has already reported on methodologies for the production of activated carbon presented in the manuscript.
Response: We thank the valuable comments and suggestions from you, which are very helpful to improve the quality of our manuscript. We have made changes accordingly and submitted the revised manuscript and supporting information file to show revisions and track changes. Enclosed below please find our point-to-point responses to the review comments.
Generally, cow manure is used as compost or solid fuel which is low in value-added with a limited consumption. To increase the consumption, it must be high in value-added as activated carbon.
This preparation method has not suggested in previous research. Generally, there are various preparation method for activated carbon in previous research. However, these method does not consider the precursor characteristics. If the properties of precursor with various precursors are not considered, it is impossible to produce high surface area of activated carbon that we aim for, as shown in Fig. R1.
Figure R1. The image of activated carbon based on dried cow manure with general method.
In this paper, we suggest the preparation methods for activated carbon based on dried cow manure with high surface area. Also, we compared the properties of activated carbon as absorbent.
Comment #1: There is no discussion of similar adsorbents reported in the literature in the introduction. There is only mention of a previous work published by the authors;
Response: We appreciate reviewer’s kind comments. As a reviewer point out, we inserted the literature and refereces about adsorbents.
On Page 3, “In addition, to adsorb odorous gases, especially ammonia, it is necessary to obtain more adsorption data on activated carbon to elucidate the influence of the surface oxides on ammonia adsorption. In previous research, the activated carbon pretreated by ozone treatment led to an increase in ammonia removal efficiency [21] and acidic surface oxides increased the adsorption capacity [22-23]. However, the relationship between the amount of oxyacid groups on the adsorbent surface and ammonia capacity has not yet been explained [24]. Therefore, this research investigates the removal of ammonia using an adsorbent, such as activated carbon.”
[21] Kim, B.J.; Park, S.J. Effects of carbonyl group formation on ammonia adsorption of porous carbon surfaces, J. Colloid Interface Sci. 2007, 311, 311-314. DOI: 10.1016/j.jcis.2007.02.059.
[22] Le Leuch, L.M.; Bandosz, T.J. The role of water and surface acidity on the reactive adsorption
of ammonia on modified activated carbons. Carbon, 2007, 45, 568-578. DOI: 10.1016/j.carbon.2006.10.016.
[23] Tamon, H.; Okazaki, M. Influence of acidic surface oxides of activated carbon on gas adsorption characteristics, Carbon, 1996, 34(6) 741-746. DOI: 10.1016/0008-6223(96)00029-2.
[24] Huang, C.-C.; Li, H.-S.; Chen, C.-H. Effect of surface acidic oxides of activated carbon on adosrption of ammonia. J. Hazard. Mater. 2008, 159(2-3), 523-527. DOI: 10.1016/j.jhazmat.2008.02.051.
Comment #2: In section 2.1, please insert the geographic coordinates referring to the sample collection;
Response: We appreciate reviewer’s kind comments. As a reviewer point out, we check the geographic coordinates and added more sentence to understand this sentence in revised manuscript as follow.
On Page 3, “The cow manure used in this study was collected from a cowshed in Anseong City (Gyeonggi Province, South Korea) and stored in Sudokwon Landfill Site (Incheon, South Korea).”
Comment #3: The figure caption of figure 1 is impressive. Please correct;
Response: We appreciate reviewer’s kind comments. As a reviewer point out, we check and exchanged to correct the Figure captions in revised manuscrip as follow,
Figure 1. The various preparation methods for activated carbon based cow manure.
Comment #4: Please insert the discussion of table 2, comparing with the literature;
Response : We appreciate reviewer’s kind comments. As a reviewer point out, we added more information to compare with the previous research in revised manuscript as follow.
On page 5, In USA [25], cow manure is 44.8% carbon and 9.4% fixed carbon whereas, in China [16], proximate and ultimate analyses of cow manure revealed 35.0% carbon and 12.4% fixed carbon content. The maximum and minimum fixed carbon contents of biomass have been recorded as 16.0% and 20.4%, respectively [30].”
Comment #5: Please Correct "optimum" to "better (or similar)". Since there is no statistical analysis to determine the optimal point of the analyses. Also, none of the results are presented in duplicate /triplicate;
Response : We appreciate reviewer’s kind comments. As a reviewer point out, we exchanged optimum to better in revised manuscript as follow.
On page 1, “The results showed that the best heat treatment temperature was 300 °C and cow manure to chemical activator ratio was 1:2.”
On page 3, “Therefore, in this study, the surface area of the activated carbon was compared and evaluated based on the augmentation processes of impurities to propose a process for the high surface area–activated carbon production from cow manure.”
On page 5, ” The ideal sample to KOH ratio was 1:2, achieving the highest surface area, while the surface area collapsed under the conditions of 1:3 or more, as found in previous research [29].”
On page 7, “Otherwise, the better acid treatment conditions are lead efficient acid removal.”
On page 8, “Therefore, the ideal temperature for cow manure heat treatment is 300 °C and the ideal cow manure to chemical activator ratio is 1:2.”
On page 10, “Figure 9. The suggested pathway to increase of surface area of precursor of activated carbon.”
On page 11, “The ideal heat treatment temperature for stabilization was determined to be 300 °C and the ideal cow manure to chemical activator ratio was 1:2.”
Comment #6: Please insert a reference to the information presented in lines 219-221;
Response : We appreciate reviewer’s kind comments. As a reviewer point out, we added reference in revised manuscript as follow.
On page 6, “However, if the concentration of acid was excessive or acid treatment is carried before the carbon structure is stable, the collapse of carbon structure is progressing, and it acts as a hindrance to the development of specifically. [29,31]”
[29] S.Y.; Kim, Lee, G.B.; Kim, J.H.; Hong, B.U; Park, J.E, Pre-Treatment Methods for Regeneration of Spent Activated Carbon, Molecules 2020, 25(19), 4561. DOI:10.3390/molecules25194561.
[31] Cevallos Toledo, R.B.; Aragon-Tobar, C.F.; Gamez, S.; Torre, E. Reactivation Process of Activated Carbons: Effect on the Mechanical and Adsorptive Properties. Molecules, 2020, 25(7), 1681. DOI: 10.3390/molecules25071681.
Comment #7: Insert SEM and TGA analyzes of activated carbon obtained in the best condition;
Response : We thank the reviewer for the valuable comments. As a reviewer point out, we added more informations such as SEM and TGA data in revised manuscript as follow.
On page 4, “To investigate surface morphogloies of activated carbons with/without chemically activated by KOH, Field-emission scanning electron microscopy (S-4300, Hitachi) was used. In addition, the thermal properties of the raw material and the carbonized sample were confirmed using a thermogravimetric analyzer (DTG-60M, Shimadzum, Japan).”
On page 7, We insert TGA data and discussed the results in revised manuscript as follows.
“Through thermogravimetric analysis were shown in Figure 3, it can be confirmed that the weight loss decreased while the fixed carbon ratio increased under the influence of heat treatment at 300℃.”
Figure 3. TGA thermograph for observation of heat treatment effect.
On page 8, we add SEM image of CM-DTH and CM-DTHCs and discussion sentences in revised manuscript as follow.
Figure 5. The scanning electron microscopy (SEM) of Activated carbon. (a) CM-DTH, (b) CM-DTHC-300(1:1), (c) CM-DTHC-300(1:2), (d) CM-DTHC-300(1:3).
“As shown in Figure 5, we confirmed that a little micropores were developed in the raw material, but the development of pores was hardly achieved. In the activated sample, the additional carbon consumption ouccurs as the amount of the chemical agent increased, and the size of the pores also increased proportionally. In the case of sample (d), mesopore or macropore mainly developed despite the increase in the number of cracks. It seems that the surface area is lowered due to the decrease in the ratio of micropore.”
Comment #8: The manuscript as a whole does not present a comparison of the analyzes with the literature, reducing its quality, despite being a work with a lot of potential;
Response : We thank the reviewer for the valuable comments. In this paper, we suggest the preparation methods for activated carbon based on dried cow manure with high surface area and the possibility as absorbent. Therefore, we clearly summarized the revised manuscript to helpful to understand the conclusion in this study.
Comment #9: Authors must review the manuscript for formal English before resubmitting the work, there are several spelling and concordance errors.
Response : We appreciate reviewer’s kind comments. We revised the manuscript based on the reviewer’s valuable comments. Also, as a reviewer point it out, we checked the English and it was also revised by “English Pre Editing service in MDPI ”.

Reviewer 2 Report
The article sent for review is very interesting and carefully prepared The article should make some corrections and additions. My suggestions are as follows:
1. Page 3 line 124, 125. t was written that the samples were annealed for 3 hours. The time the sample remains in the sample - 1 hour. Growth rate of 5oC / min. Are these data correct for all temperatures: 300,500 and 700oC? The time to heat up to 300oC is 60 minutes, and to 700oC - 140 minutes. Is the company in the article in time?
2. Page 4 line 134. Does the given time of 3 h apply to the entire process or to the reaction at 850oC.
3. Page 4 line 147-148. The authors cited earlier studies. There is no literature source.
4. Report how to calculate recovery rates (line 194)
5. Page 6 line 205. How was the 23% score calculated? Comparing fixed carbon from table 4 to trials with sample 5, a different result (24.2-17.8)/17.8 = 36).
6. Page 6 line 211. Is the 30% value correctly calculated? Wouldn't I have to calculate it as follows: (1034-642)/642 ???
7. Page 7 line 246,24. Are the given results (55, 85, 95%) calculated correctly? Is the result in table 7 (volatile 6.6%) correct? When the reaction took place at 700°C for samples 1, 2 and 3 the results were higher than for sample 0. So the volatile substances content increased. Were these results compared to Sample 0? The article does not explain this well.
8. Information on the ammonia adsorption mechanism should be expanded.
9. The summary contains the information about the toluene adsorption test for the first time. Were there no mistakes?
Author Response
Responses to Referee 2
General Comment: The article sent for review is very interesting and carefully prepared The article should make some corrections and additions. My suggestions are as follows:
Response: We thank the valuable comments and suggestions from you, which are very helpful to improve the quality of our manuscript. We have made changes accordingly and submitted the revised manuscript and supporting information file to show revisions and track changes. Enclosed below please find our point-to-point responses to the review comments. We also checked the English and it was also revised by “English Pre Editing service in MDPI ”.
Comment #1: Page 3 line 124, 125. t was written that the samples were annealed for 3 hours. The time the sample remains in the sample - 1 hour. Growth rate of 5oC / min. Are these data correct for all temperatures: 300,500 and 700oC? The time to heat up to 300oC is 60 minutes, and to 700oC - 140 minutes. Is the company in the article in time?
Response: We appreciate reviewer’s kind comments. As a reviewer point out, we corrected this sentence in reivsed manuscript as follow;
On page 4, “The temperatures for heat treatment were within the range of the chemical activation temperature. The samples were denoted as CM-DTH. The samples were placed in alumina crucibles and put into a tubular furnace under N2 flow. The N2 was purged at 100 ml/min. The temperature was increased to 300, 500, and 700 °C at a rate of 5 °C/min and maintained for 3 h to obtain CM-DTH-300, CM-DTH-500, and CM-DTH-700, respectively.
Comment #2: Page 4 line 134. Does the given time of 3 h apply to the entire process or to the reaction at 850oC.
Response: We appreciate reviewer’s kind comments. As a reviewer point out, we check and exchange all sentence in experiment section in revised manuscrip as follow;
On page 4, “The temperature increased at 850 °C at a rate of 5 °C/min and then maintained for 3 h.”
Comment #3: Page 4 line 147-148. The authors cited earlier studies. There is no literature source.
Response: We appreciate reviewer’s kind comments. As a reviewer point out, we added the reference in revised manuscript as follows;
On page 4, “An elemental analysis (EA) was conducted using an elemental analyzer (Flash EA 1112, Thermo Scientific, Milan, Italy) by following the procedures described in a previous study [28].”
Comment #4: Report how to calculate recovery rates (line 194)
Response: We appreciate reviewer’s kind comments. As a reviewer point out, we exchange “recovery rate” to “yield” and added the calcuate equation to understand in revised manuscrip as follows;
On page 1, “. The heat treatment stabilization process increases the fixed carbon ratio and the solid yield, and the acid wash process removes substances that restrain the increase in surface area.”
On page 5, “Solid yields were calculated the using the following equation and compared.
Solid Yield (%)= (Output solid as activated carbon )/(Input solid as precursor) x 100
The solid yields were between 8–32% due to the low fixed carbon tents and the solid yield tended to decrease with an increase in surface area.”
On Page 11, “Consequently, direct chemical activation of cow manure with a low fixed carbon ratio resulted in low surface area and solid yield. Additionally, stabilization at high temperatures with unstable carbon bonds decreased the surface area and solid yield. The surface area and solid yield increased only when carbon bonds were stabilized before chemical activation. However, excessive input of chemical activators when carbon bonds of carbon precursors are weak decreases the surface area and solid yield.”
Comment #5: Page 6 line 205. How was the 23% score calculated? Comparing fixed carbon from table 4 to trials with sample 5, a different result (24.2-17.8)/17.8 = 36).
Response: We appreciate reviewer’s kind comments. As a results, we check all errors and exchange all sentence in revised manuscrip as follows;
On page 6, “ From the results of HTC, the fixed carbon content of CM-DT was increased by about 36% compared to that of CM-D, as shown in Table 4.”
Comment #6: Page 6 line 211. Is the 30% value correctly calculated? Wouldn't I have to calculate it as follows: (1034-642)/642 ???
Response: We appreciate reviewer’s kind comments. As a results, we check all errors and exchange all sentence in revised manuscrip as follows;
On page 6, “The surface area of CM-DTC was generally higher than that of CM-DC. Especially, a sample to KOH ratio ratio of 1:2 showed a 91% increase in surface area. The fixed carbon content of CM-DT was 36% higher than that of CM-D and that of CM-DTC was generally higher than that of CM-DC, except for a 1:1 sample to KOH ratio. Also, CM-DTC had a lower ash content than CM-DC.”
On page 7, “Stabilization through heat treatment was performed, expecting an increase in fixed carbon ratio and surface area of CM-DT, as HTC only resulted in a 10% (35.4% of CM-DT – 24.2% of CM-D) increased grade in fixed carbon ratio.”
Comment #7: Page 7 line 246,24. Are the given results (55, 85, 95%) calculated correctly? Is the result in table 7 (volatile 6.6%) correct? When the reaction took place at 700°C for samples 1, 2 and 3 the results were higher than for sample 0. So the volatile substances content increased. Were these results compared to Sample 0? The article does not explain this well.
Response: We appreciate reviewer’s kind comments. As a results, we add more sentence about CM-DTH and correct all calcuated numbers in revised manuscrip as follows;
On page 8, “In the case of CM-DTH, the fixed carbon content generally increased with increasing pretreatment temperature. Also, the volatile matter of CM-DTHs decreased with increasing temperature: 32% for CM-DTH-300, 69% for CM-DTH-500, and 90% for CM-DTH-700 compared to CM-DT. However, CM-DTHs had low surface areas, even with hydrothermal and pre heat treatments.”
The increase in the volatile is due to the decrease in the content of fixed carbon and the increase in volatile matters when high pre heat temperature as 700C. It is determined that the volatile content is increased from CM-DTH to CM-DTHC, and this collapse phenomenon occors when high concentration of activator or strong acid. [31] Therefore, this phenomenological explaintion is added to this sentence in revised manuscrip as follows.
On page 8, “In the case of CM-DTHC-700, the volatile content was high and increased with the activator ratio. There is a possiblity of collapse of carbon structre owing to the high pretreatment temperature.”
Comment #8: Information on the ammonia adsorption mechanism should be expanded.
Response : Thank you for your kindness response. In previous research, they proposed that the adsorption mechanism of the rice husk-based adsorbents and followed the pseudo-second-order model. We inset the ammonia adsorption mechanism in revised manuscript as follow;
On Page 11, “Previous research determined that the adsorption process of ammonia onto activated carbon follows the pseudo-second-order model [39]. The pseudo-second order kinetic model suggests that chemical adsorption is the dominant mechanism. The surface area related with porous structures, chemisorption based on Lewis acid-base reaction.”
Comment #9: The summary contains the information about the toluene adsorption test for the first time. Were there no mistakes?
Response : Thank you for your kindness response. We had some mistake sentences in maunuscript, so we checked all sentences in revised manuscript as follow;
On page 11, “This study evaluated the potential of cow manure as a precursor to high surface area–activated carbon and compared its performance to commercial counterparts using the ammonia adsorption test.”

Round 2
Reviewer 1 Report
The authors improved the quality of the manuscript significantly. Now I recommend the publication of this work.